# A Case Report of a Botulism Outbreak in Beef Cattle Due to the Contamination of Wheat by a Roaming Cat Carcass: From the Suspicion to the Management of the Outbreak

**DOI:** 10.3390/ani9121025

**Published:** 2019-11-25

**Authors:** Caroline Le Maréchal, Olivier Hulin, Sabrina Macé, Cécile Chuzeville, Sandra Rouxel, Typhaine Poëzevara, Christelle Mazuet, Françoise Pozet, Eric Sellal, Laure Martin, Alain Viry, Christine Rubbens, Marianne Chemaly

**Affiliations:** 1ANSES, Laboratoire de Ploufragan—Plouzané, Unité Hygiène et Qualité des Produits Avicoles et Porcins, BP 53, 22440 Ploufragan, France; Sabrina.MACE@adria.tm.fr (S.M.); Sandra.ROUXEL@anses.fr (S.R.); Typhaine.POEZEVARA@anses.fr (T.P.); laure.martin@aol.com (L.M.); Marianne.CHEMALY@anses.fr (M.C.); 2Cabinet Vétérinaire de Bletterans, 16, Rue de la Demi-Lune, 39140 Bletterans, France; veto.bletterans@gmail.com; 3GDS 71, Loche, 99 Rue des Grands Crus, 71000 Mâcon, France; cecile.chuzeville.gds71@reseaugds.com; 4National Reference Center for Anaerobic Bacteria and Botulism, Pasteur Institute, 25-28 rue du Docteur Roux, 75724 Paris, France; christelle.mazuet@pasteur.fr; 5Laboratoire Départemental d’Analyses du Jura, 59 Rue du Vieil Hôpital, 39800 Poligny, France; fpozet@jura.fr (F.P.); aviry@jura.fr (A.V.); 6Agrivalys 71, 267 Rue des Épinoches, 71000 Mâcon, France; direction@agrivalys71.fr; 7Direction Départementale de la Protection des Populations de la Saône et Loire, 71 Rue Jean Macé, 71000 Mâcon, France; christine.rubbens@saone-et-loire.gouv.fr

**Keywords:** cattle botulism, cat carcass, PCR, wheat, *Clostridium botulinum*, feed contamination

## Abstract

**Simple Summary:**

This study presents in detail a botulism outbreak in a beef cattle farm where the source of contamination was identified as a carcass of a roaming cat that had contaminated stored feed and initiated the botulism outbreak. In this case report, we present how the diagnosis was performed by describing the clinical signs and the analyses that were conducted; how the source of contamination was identified by analyzing samples collected on the farm, and finally, how the outbreak was managed so as to prevent recurrence of the disease and persistence of the contamination in the farm.

**Abstract:**

We report a botulism outbreak in Charolais cattle fed with wheat flour contaminated by *Clostridium botulinum* type C and the management of the outbreak at each step from the clinical suspicion to the cleaning and disinfection operations. Diagnosis was based on typical suggestive clinical signs and detection of *C. botulinum* type C using real-time PCR in samples collected from three young affected bulls. All young exposed bulls and cows (18 animals) eventually died, but three young bulls and one cow were recovering when it was decided to euthanize them. *C. botulinum* type C was detected in the liver of these four animals. Analysis of the ration components demonstrated that wheat flour, wheat, and the mill used to make flour were positive for *C. botulinum* type C. A dead cat positive for *C. botulinum* type C was discovered in the silo where wheat grain was stored and was considered the source of contamination. The cat’s entire body was found mummified, well preserved, and not rotting in the silo. Specific measures, in particular, vaccination of the rest of the herd and cleaning and disinfection operations, were implemented to prevent any recurrence of the outbreak. The presence of wild animal carcasses in feed harboring anaerobic conditions like silage, in particular during harvesting, are known to be at risk for the initiation of a botulism outbreak. This outbreak is a reminder that the presence of an animal carcass in feed, regardless of the kind of feed and whenever the contamination occurs, either during harvesting or storage, is sufficient to induce a botulism outbreak.

## 1. Introduction

Botulism is a severe neuroparalytic disease that affects humans, non-human mammals, birds, and fish. It is caused by botulinum neurotoxins (BoNTs) produced by BoNT-producing clostridia, in particular, *C. botulinum*. BoNTs inhibit acetylcholine release in cholinergic nerve endings, resulting in muscle paralysis [1].

BoNTs are divided into seven serologically distinct types, A–G, and around 40 subtypes have been described up to now [2]. Human botulism is associated with BoNT types A, B, E, and F, while animal botulism is mostly associated with BoNT types C, D, C/D, and D/C [3]. Types A, B, C, D, and D/C have been shown to produce disease in cattle [1,4], while type D/C is the most common current BoNT associated with cattle botulism in Europe [3,5].

Botulism intoxication occurs either by the ingestion of preformed toxins or wound or intestinal toxicoinfection. Ingestion of preformed BoNTs has been assumed to be the main route of contamination in cattle for a long time [6]. It has also been suggested that bovine botulism results from the ingestion of *C. botulinum* spores that germinate, multiply, and produce BoNT in the bovine gastrointestinal tract [7]. The most common sources of contamination for cattle botulism reported in the literature are ingestion of water or silage contaminated by small mammal or bird carcasses [8,9], poultry litter [10,11], or ingestion of insufficiently acidified silage contaminated by *C. botulinum* spores [12].

Animal botulism has been reported to be a concern in Europe for the last decade, with an increase in the number of reported outbreaks [13]. Many cattle botulism outbreaks have also been reported outside of Europe, with some cases involving hundreds of animals [8,12,14]. This induces economic losses that can be extremely high for the farmer and create challenges in terms of managing the outbreak, preventing cross-contamination, and avoiding recurrence of the disease.

Here, we describe a cattle outbreak in beef cattle, the approach to confirm the diagnosis, the identification of the source of the outbreak, and its management, in particular, the cleaning and disinfection operations.

## 2. Materials and Methods

### 2.1. Case History

An outbreak of botulism occurred in the spring of 2018 on a farm with Charolais breeder fattener operations in Saône-et-Loire, Eastern France. A backyard pen with 4 laying hens was also present on the farm near the house, and 25 broilers were introduced at the beginning of June. Figure 1 represents the different barns on the farm.

Twenty-four young bulls were raised in barn number 6, and 4 fattening cows were housed in barn number 3. The rest of the herd was grazing in outside pastures at the time of the outbreak. Figure 2 presents the chronology of events.

Of the 24 young bulls, 6 were slaughtered in April without any clinical signs. One cow was slaughtered on 28 May without any clinical signs. Of the remaining 18 young bulls, 2 died suddenly on 9 and 23 April. No investigation was performed for these animals. From 19 May to 4 June, a total of 13 young Charolais bulls and 2 cows died after having developed signs of apathy, anorexia, weakness, paralysis, lateral recumbency, and flaccid paralysis of the tongue in some animals (Figure 3).

All animals presented constipation and polypnea with contraction of abdominal muscles after blowing air out. Tremors were also observed as well as increased salivation and mastication difficulties for many animals. No fever was observed. Different levels of clinical signs were observed among animals and ranged from subacute to peracute. Two animals, in fact, died within a few hours on 20 May, while 3 young bulls and 1 cow presented clinical signs but slowly recovered. Considering the context and presence of low persistent clinical signs, the decision was taken to euthanize these animals. Other animals were sick from 2 to 7 days before death.

Botulism was suspected on 31 May. Animal botulism is a notifiable disease in France [15]. A prefectural monitoring decree (*Arrêté préfectoral de mise sous surveillance*) was therefore implemented by the local veterinary services (*Direction départementale de la protection des populations de la Saône-et-Loire*) on that day.

### 2.2. Sampling of Animal Biological Material

The veterinary officer necropsied 2 young bulls on 1 June, 36 h and 24 h after natural death (young bull no. 2 and 3, visible in Figure 2). A symptomatic young bull was euthanized and immediately sampled by the veterinary officer on 4 June (young bull no. 1). The liver, gall bladder, rectal contents, ruminal contents, feces, hilum, and intestines were frozen before being submitted for *C. botulinum* examination.

Fecal samples were collected on 18 and 25 June from the 4 animals that recovered from clinical signs. After euthanasia, the liver and ruminal contents were also collected. All samples were frozen before being submitted for *C. botulinum* examination.

Ceca from broilers and laying hens and feces from the farm dog were also collected on 25 June. Feces from the farmer and veterinary officer were collected on 23 July.

A cat carcass was found in the wheat silo (Figure 2). Hairs, bones from the ribcage, and skin were collected from the carcass, and the inside of the carcass was swabbed.

### 2.3. Epidemiological Investigation and Environmental Sample Collection

Investigations were performed on 4, 12, 18, 25 June, 3 August, and 4 September to identify the source of contamination and monitor *C. botulinum* dissemination on the farm.

Samples of water, cereal flour, wheat stored in a silo, barley stored in another silo, grass silage, and manure collected in house number 6 and 4 were collected.

Swabs and boot-swabs (Sodibox, Nevez, France) were collected in the farm barns (floors and walls), in silos, in the mill used to produce flour from barley and wheat, and the hay.

### 2.4. Culture Conditions and DNA Extraction

#### 2.4.1. For Animal Biological Materials for Diagnostic Purposes

Two enrichment protocols were used for bovine samples. For the first protocol, 1 gram of each matrix was individually diluted ten-fold in pre-reduced Fortified-Cooked Meat Medium (F-CMM) (pre-incubated at 70 °C) and incubated for 10 min at 70 °C before being cooled down in cold water. For the second protocol, 25 grams of each matrix were individually diluted in pre-reduced Trypticase Peptone-Glucose-Yeast extract broth (TPGY) (by boiling for 10 min before use and cooling down to room temperature) with a 1:10 dilution.

After incubation at 37 °C ± 2 °C under anaerobic conditions (A35, Don Whitley distributed by Biomérieux, Bruz, France) for at least 22 h, 1 mL of each enrichment broth was collected. Cells were pelleted by centrifugation and subjected to DNA extraction using the QiaAmp DNA mini-kit (Qiagen, Courtaboeuf, France) according to the manufacturer’s instructions for all samples.

#### 2.4.2. For Human Fecal Samples

Detection of *C. botulinum* in human fecal samples was based on culturing samples in freshly prepared F-CMM at 37 °C under anaerobic conditions. After a 48-h period of incubation, 1 mL of enrichment culture was collected, and DNA was extracted from the pellet cells using a QIAamp DNA Stool Kit (Qiagen, Courtaboeuf, France) according to the manufacturer’s instructions.

#### 2.4.3. For Environmental Samples

Swabs and boot swabs were diluted in 250 mL pre-reduced TPGY. Ceca, dog feces, and carcass samples were individually diluted in pre-reduced TPGY with a 1:10 dilution. Water (100 mL) was diluted in pre-reduced TPGY 2 times concentrated with a 1:2 dilution. Silage, cereal flour, wheat, barley, and manure were 2 times diluted in pre-reduced TPGY, and 50 mL of this diluted solution was 5 times diluted so as to analyze a final weight of 25 g for each matrix according to the recommendations of the NF EN ISO 6887-6 Standard. All samples were homogenized for 15 s using a Pulsifier^®^ (Microgen, Surrey, UK) and incubated for at least 4 days at 37 °C ± 2 °C in an anaerobic station (A35, Don Whitley distributed by Biomérieux, Bruz, France) filled with anaerobic gas (10% H_2_, 10% CO_2_, 80% N_2_). After incubation, 1 mL of each enrichment broth was collected. Cells were pelleted by centrifugation and subjected to DNA extraction using either the Powersoil DNA isolation kit (Qiagen, Courtaboeuf, France) or the Nucleospin Soil kit (Macherey, Duren, Germany) depending on matrices [16] according to the manufacturer’s instructions.

### 2.5. Real-Time PCR

Except for human fecal samples, the real-time PCR, primers, and probes used in this study have been described previously [3,11]. PerfeCTa qPCR ToughMix (VWR, Briare, France) was used instead of iQ Supermix (Bio-Rad, Marne-La-Coquette, France). A sample was considered positive when characteristic amplification was detected.

Detection of the *bont/C* gene in human fecal samples was performed by SYBR green real-time PCR (iQ SYBR green Supermix, Bio-Rad, Marne-La-Coquette, France) according to the protocol previously described [17] using primers P1652 (5′-GGCACAAGAAGGATTTGGTG-3′) and P1653 (5′-TTGGATCCATGCAAAATTCA-3′).

### 2.6. Detection of BoNT in Human Fecal Samples

Detection of BoNT in the 96-h enrichment culture of human fecal samples was based on a mouse lethality assay. The tests were performed in accordance with European Directive 2010/63/EU on the protection of animals used for scientific purposes (laboratory animal use agreement no. 2013-0116). One milliliter of enrichment broth was collected, centrifuged, filtered, and 5 times diluted in 50 mM phosphate buffer (pH 6.5) containing 1% gelatin. A volume of 0.5 mL was injected intraperitoneally into Swiss mice weighing 20–22 g (Charles River Laboratories, l’Arbresle, France). The mice were observed for up to 4 days for the presence of typical clinical signs (pinching of the waist, labored breathing, and paresis) and euthanized immediately after observation of such signs.

## 3. Results

### 3.1. Laboratory Confirmation of the Type C Botulism Outbreak

Necropsy of animals did not enable detection of any significant gross lesions except in one young bull that had thoracic purulent lesions with fistula. Samples were collected from seven symptomatic animals, including four that were recovering. Results are presented in Table 2
*C. botulinum* type C was detected in all tested animals.

When considering both protocols (enrichment in TPGY and F-CMM) at the same time, all samples collected from dead animals were positive, while only three samples out of six were positive on the freshly euthanized one.

Surprisingly, *C. botulinum* type C was detected in the four animals that were recovering, notably in the liver, even 21 days after the last death reported on the farm and one month after the end of exposure to contaminated feed.

No *C. botulinum* type C was detected in human fecal samples collected from the farmer and the veterinarian.

### 3.2. Detection of *C. Botulinum* in Samples Collected on the Farm

Results on the detection of *C. botulinum* on samples collected on the farm are presented in Table 2.

Young bulls and fattening cows had the same ration composed of hay, wheat, and maize flour produced by the farmer and commercial rapeseed cake. Drinking water was tap water. Samples of flour, rapeseed cake, and water were first analyzed. Flour was positive with an early PCR signal (mean Ct of 25), and one water sample collected in one of the drinking trough in house no. 6 was positive with a late signal (mean Ct of 37.5). Considering these results, wheat was collected at the top of the silo and at the exit (just before the entrance of the mill). Barley and grass silage were also analyzed. The inside of the mill was swabbed. Both wheat samples and swabs collected in the mill were positive for *C. botulinum* type C, showing that the source of the outbreak was contaminated wheat.

The silo containing wheat was then emptied. A dry mummified cat carcass was found at the bottom of the silo (Figure 4) on 23 June. It was positive for *C. botulinum* type C.

Swabs and boot-swabs were performed to evaluate the dissemination of *C. botulinum* type C on the farm. A swab inside the mill was positive, even after the first cleaning and disinfection operations. Only a few other samples were positive.

### 3.3. Management of the Outbreak

Considering the different positive samples collected from this farm, a by-law declaring infection (*Arrêté préfectoral de declaration d’infection*) was implemented by the local veterinary services. Several control measures were implemented: an inventory of all animals on the farm, a ban on the movement of animals, manure, feed, hay or straw, meat products, and equipment (except for specific exemptions that would be issued by the local veterinary services), as well as a ban on the presence of pets on the farm, isolation of diseased animals from healthy ones, vaccination of cattle, carcass collection for rendering, reinforced biosecurity measures, and disinfection operations (restrictions on the entry of personnel, specific clothes and shoes, hand-washing, disinfection of the equipment, the house, and the surrounding area). These control measures continued until three weeks after the last detected signs and the disinfection operation.

The cow slaughtered on 28 May was placed under quarantine due to the mortality context on the farm, and its marketing was authorized when botulism type C was confirmed. The cow was healthy when sent to the slaughterhouse, and according to the opinion published in 2002 by AFSSA [18], risks for humans were considered low to negligible regarding botulism type C. However, it was forbidden to use vacuum-packaging for meat.

On 25 May, the feed ration was changed: distribution of cereal flour and concentrate was stopped, and a remaining batch of hay from 2015 was distributed instead of the current one. A water container was set up near the animals to allow them to drink more easily.

Feces from animals no. 4 to 7 (as shown in Table 2) were positive for *C. botulinum* type C. Even though they were recovering, they could not be considered healthy and were not considered suitable for slaughtering. Moreover, young bulls were supposed to be culled in June 2018. As the time for full recovery and the possibility of relapse were unknown, it was considered too risky from an economic point of view to continue raising these animals. It was, therefore, decided to euthanize them. Finally, these four animals represented a source of contamination of the environment through the dissemination of *C. botulinum*.

All cattle that were on pasture during the outbreak were vaccinated as a preventive measure using Ultravac Botulinum^®^ (Zoetis, Paris, France). Two doses were injected within a one-month interval.

It was also decided to euthanize poultry from the farm to avoid the persistence of *C. botulinum* on the farm or dissemination of spores and to be able to properly conduct a total clean-out of the farm.

Contaminated wheat, barley, manure from all houses, and hay that was stored near young sick bulls were burnt on a plot of land far from any house on 2 July under control of the local fire brigade.

Then, cleaning and disinfection operations were carried out by a specialized private company. Equipment was disassembled as far as possible so as to facilitate the cleaning process. A high-pressure pump combined with foaming detergent was used for cleaning operations. After cleaning, disinfection was performed using 24% liquid formaldehyde solution diluted to 5% applied with a foam gun (94 L of initial formaldehyde solution was used on the whole farm). After these initial operations, one swab collected on the mill and bushel was still positive. Additional cleaning and disinfection operations were implemented to remove *C. botulinum* spores. Tubulars and bushels were again fully disassembled. It was not feasible to disassemble the mill, but it was disinfected using a blowtorch and then using the method described above. Tubulars were soaked in a formaldehyde bath (a 24% formaldehyde solution diluted to 5% was used). Samples were collected before reassembly to assess contamination. *C. botulinum* was not detected after this second operation.

## 4. Discussion

Feed presenting anaerobic conditions, like silage or wrapped grass, is known to be at risk for *C. botulinum* growth and BoNT production [19]. This case report shows that dry matrices, including grains, are also to be considered as potential sources for bovine outbreaks.

Most of the cattle outbreaks reported in the literature are dairy cattle [8,10,11,12,14,20,21,22,23,24,25]. This outbreak emphasizes that beef cattle can also be affected by botulism, even in small farms and not only in feedlots [26,27].

The diagnosis was based on the detection of *C. botulinum* using real-time PCR in various samples collected from three animals. The gold standard for botulism diagnosis is to demonstrate the presence of BoNTs using a mouse bioassay in serum or samples collected on symptomatic animals. However, the detection of BoNT in bovine samples is complex: cattle are, in fact, 12.88 times more sensitive to BoNT type C than mice on a per kilogram weight basis with the median toxic dose of BoNT C for cattle being 0.388 ng/kg [4]. As a result, the available methods are not sufficiently sensitive to detect BoNTs in bovine samples. Demonstrating BoNTs or BoNT-producing clostridia in samples collected on cattle, or in their environment or feed combined with clinical signs evocative of botulism, is considered valuable for botulism diagnosis [19]. This was the approach used here to diagnose cattle botulism and has been successfully reported several times in the literature [10,12,14,26].

Two enrichment protocols were used in our study, and they appear to be complementary for the detection of *C. botulinum* type C (Table 2): Protocol 1 enabled better detection in feces (6/8) than Protocol 2 (2/8), while Protocol 2 allowed for better detection in the liver (6/7 against 2/7 for protocol 1). This result might suggest that *C. botulinum* is mostly found as spores (resistant to thermal treatment at 70 °C) in feces and as vegetative cells (sensitive to thermal treatment) in the liver.

The BoNT mainly responsible for bovine botulism in Europe and Japan is type D/C [3,28]. Bovine type C outbreaks are more rarely reported [3,28,29] even though they do occur [5]. They are, however, commonly encountered in Australia and South Africa, where bovine and avian husbandry are mixed [29]. Several type C outbreaks of bovine botulism due to contamination of feed with a carcass, especially a cat carcass, have been reported in the literature [8,14,21,22,23,24,30,31]. Contaminated hay or silage was involved each time. Here, in this case report, wheat that was dry and not processed to harbor anaerobic conditions unlike silage or wrapped grass was the contaminated feed. The entire cat carcass was found in the silo (Figure 4), indicating that it was not included in wheat during harvesting, in which case, pieces of the animal would have been found. It seems more likely that this cat fell into the wheat during transport from field to farm or during storage. Straw was stored near the silo, and the cat could have fallen down into the silo and then been unable to get out. It has also been reported that cats are sensitive to type C botulism when they are fed with contaminated carrion [32]. Therefore, it can also be hypothesized that botulism could be responsible for cat death.

Few studies have been conducted to evaluate asymptomatic carriage of *C. botulinum* in pets or wild animals, especially near farms and their potential role in the initiation of animal botulism outbreaks. The presence of BoNT C has been demonstrated in feces collected from cats with feline dysautonomia and BoNT C *C. botulinum* in dried cat feed [33]. The cat found here did not belong to the farmer, his family, or the neighbors. Its origin was not identified, and it could be considered as a roaming cat. Its diet was, therefore, unknown. It could be assumed that free-roaming cats may be exposed to *C. botulinum* type C through hunting.

Both wheat samples collected at the top of the silo or near the entrance of the mill were positive, showing massive contamination of the silo. Like the use of mixer wagons [14,19], which is known to induce a homogenized distribution of BoNT in feed, the mill may also have facilitated massive contamination of flour made from wheat.

Considering the presence of the cat carcass in the wheat silo, the carcass was considered to be the source of the outbreak. The presence of *C. botulinum* type C in water (sampled from 4 June) must result from secondary contamination, either by contaminated animals or during feed distribution.

The time period in which the animals were sick was 26 days. Even after ration modification, thus stopping animal exposure to the source of BoNTs and BoNT-producing clostridia, animals exhibited clinical signs. In previously reported outbreaks, similar incubation times of 19 days, 16 days, and 18 days have been reported [8,10,14]. Variable symptom severity was also observed during the outbreak: from a few hours to several days before death. Variable severity has also been described during another type C botulism outbreak [20]. Moreover, the fact that four animals began to recover confirmed biphasic bovine botulism outbreaks that have been reported elsewhere: an acute and rapid phase with the onset of clinical signs in less than 96 h and a delayed phase with the disease only two weeks after exposure to contaminated feed [34]. This latter phase could be due to toxicoinfection: exposure to a low amount of BoNT might induce the debilitation of animals and provide suitable conditions for in vivo BoNT production [34]. Here, *C. botulinum* was detected in various samples collected on animals. Three out of the four animals that were recovering were also the last to become symptomatic. Vegetative cells were found in the livers of the four animals, ruminal contents of two animals, and feces of one of them, supporting the hypothesis of in vivo BoNT production. Prolonged exposure to sub-lethal amounts of BoNT C may explain the clinical signs observed in the four animals [20]. *C. botulinum* spores are known to be able to survive many years in the environment [25] and are able to persist several months in manure [35]. In this study, the feces of the analyzed animals were positive for *C. botulinum* type C, indicating that manure was probably contaminated. Spreading of the bacterium may induce a risk of soil, grass, or crop contamination. Dissemination of spores during manure spreading could also represent a source of contamination in the environment and a risk for other farms in the neighborhood [25]. Wheat grains stored in the silo were also contaminated. Burning of contaminated matrices is the most efficient available solution to manage such matrices and to prevent any contamination or spore dissemination in the environment [18]. Here, ashes collected at the beginning of September did not show *C. botulinum*.

*C. botulinum* spores are highly resistant, and cleaning and disinfection operations are difficult. The mill and bushel were cleaned and decontaminated twice as *C. botulinum* was still detected after the first operations, even though formaldehyde solution was used. Few studies are available regarding the efficacy of biocides on *C. botulinum*, especially on group III, but formaldehyde has been reported to be effective for decontamination after a botulism outbreak [18,36,37,38]. The difficulties encountered to properly clean the mill and bushel may also explain the failure of the first cleaning and disinfection operations. A combination of physical (heat) and chemical (formaldehyde solution) operations for the second disinfection resulted in the absence of detection of *C. botulinum*. Conducting cleaning and disinfection operations in a cattle farm is really challenging as barns and livestock management are not designed for this purpose. This investigation provides useful information for the implementation of efficient measures to get rid of spores and prevent a recurrence of the outbreak.

Contamination of the farmer and the veterinarian was evaluated by analyzing their fecal specimens. *C. botulinum* type C was not detected, although they were in contact with sick animals and contaminated matrices. Such investigations have hardly ever been conducted up to now [39] although they are of main importance to evaluate worker’s exposure during outbreaks.

## 5. Conclusions

This type C botulism outbreak induced the loss of 18 cattle representing 100% mortality among animals fed with contaminated flour. A cat carcass was identified as the source of contamination of wheat grains that were milled before being used to feed animals.

This outbreak illustrates the importance of properly storing feed to avoid feedborne botulism and how crucial it is to avoid the presence of any carcasses to prevent the onset of a botulism outbreak. It also demonstrates that all feed components in rations must be considered when looking for the source of an outbreak and that silage, wrapped grass, or haylage are not the only matrices responsible for cattle botulism outbreaks. Real-time PCR for the detection of *C. botulinum* combined with enrichment protocols was found to be valuable for laboratory confirmation of cattle botulism, identification of the source of the outbreak, and monitoring of cleaning and disinfection operations. All these steps together were crucial to properly manage this outbreak and brought new insights that provide a better understanding of this disease and crucial new elements for the management of cattle botulism outbreaks regarding the diagnosis, identification of the source of contamination, the management of animals, and cleaning and disinfection operations.

## Figures and Tables

**Figure 1 animals-09-01025-f001:**
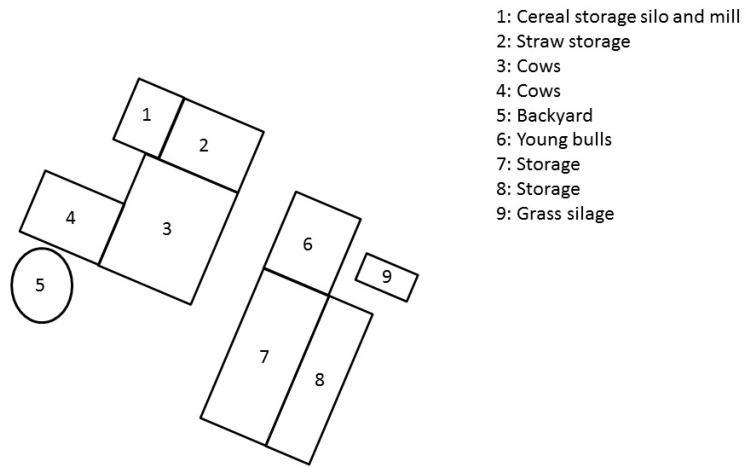
Layout of the Charolais cattle farm site that experienced a type C botulism outbreak in France in spring 2018.

**Figure 2 animals-09-01025-f002:**
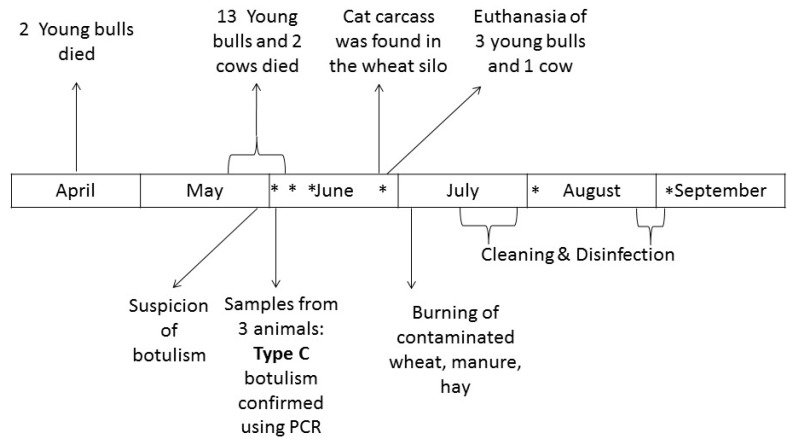
Chronology of events. * Environmental samples were collected on the farm (see Table 1 for more details).

**Figure 3 animals-09-01025-f003:**
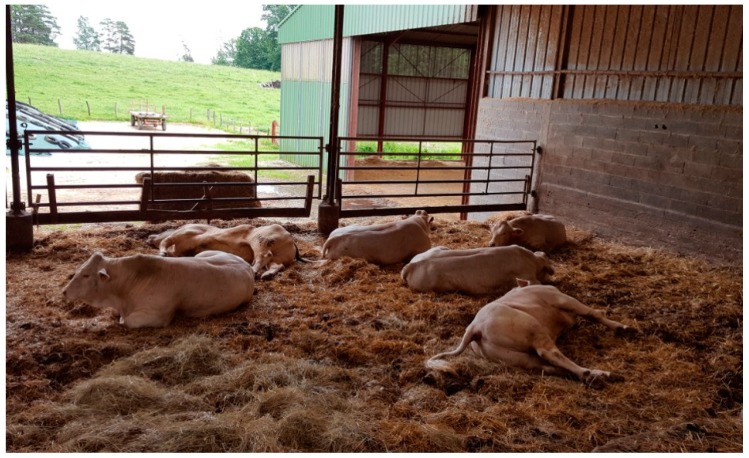
Young bulls with clinical signs (in house 6) on 31 May. One young bull (no.2 in Table 2) died just before the veterinary officer’s visit (top left of the picture), and a second one (no.3 in Table 2) died just after the departure of the veterinarian officer (bottom right of the picture). Note recumbency, apathy, and weakness.

**Figure 4 animals-09-01025-f004:**
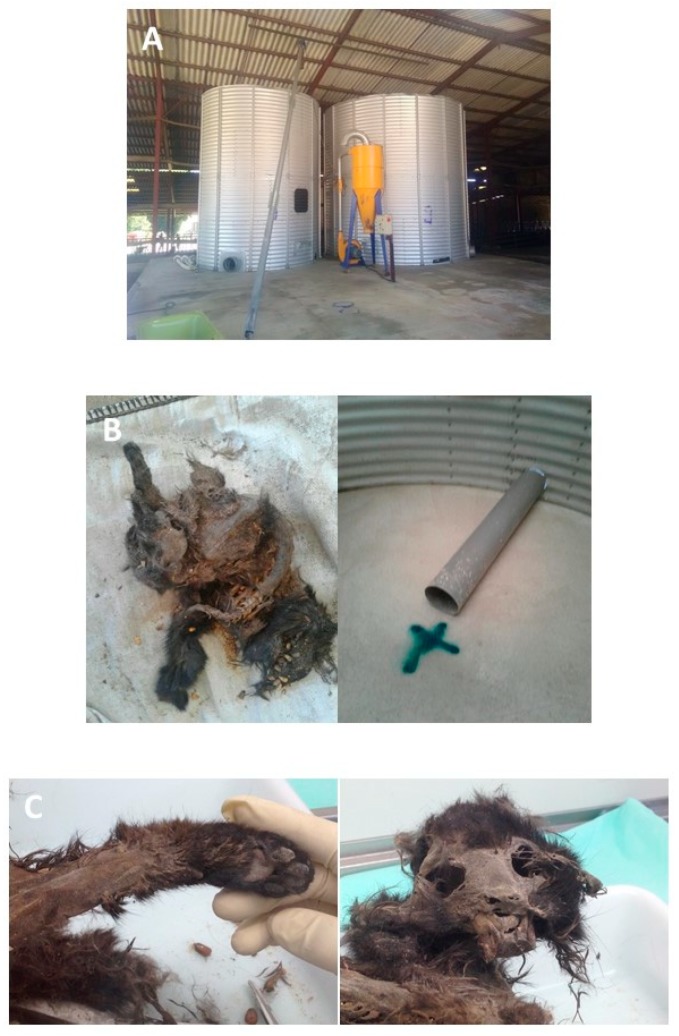
Carcass of a cat found in the wheat silo (wheat that was used to make flour distributed to animals). (**A**) Silo used for wheat and barley storage, mill, and bushel after cleaning and disinfection. (**B**) On the left, the cat carcass just after its discovery and on the right its exact position in the wheat silo. (**C**) Two pictures during the carcass examination at the laboratory before sampling for *C*. *botulinum* type C gene detection.

**Table 1 animals-09-01025-t001:** Detection of *C. botulinum* type C using real-time PCR.

House No.	Sample	Visit 1	Visit 2	Visit 3	Visit 4	Visit 5 *	Visit 6 *
		4 June 2018	12 June 2018	18 June 2018	25 June 2018	3 August 2018	4 September 2018
1	Cereal flour	C	-	-	-	-	-
Container	Rapeseed cake	ND	-	-	-	-	-
3	Water from the drinking trough	ND	-	-	-	-	-
6	Water from the drinking trough	C	-	-	-	-	-
6	Water from the drinking trough	ND	-	-	-	-	-
6	Swab in front of young bull box	-	ND	-	-	-	-
6	Swab at the back of the young bull box	-	ND	-	-	-	-
2	Bootswab in front of bales of straw	-	ND	-	-	-	-
3	Swab in the unwinter box	-	C	-	-	-	-
3	Swab on hay batch 1	-	ND	-	-	-	-
3	Swab on hay batch 2	-	ND	-	-	-	-
1	Swab on big bags	-	ND	-	-	-	-
1	Boot-swab in front of wheat silo	-	ND	-	-	-	-
1	Swab on the outside of the wheat silo	-	ND	-	-	-	-
1	Swab on the outside of the mill	-	ND	-	-	-	-
6	Boot-swab on the feeding alley	-	ND	-	-	-	-
1	Wheat collected at the top of the silo	-	C	-	-	-	-
1	Wheat collected at the bottom of the silo	-	C	-	-	-	-
1	Barley	-	ND	-	-	-	-
1	Grass silage	-	ND	-	-	-	-
1	Swab inside the mill	-	-	C	-	-	-
2	Swab at the back of the wheat silo	-	-	ND	-	-	-
6	Manure	-	-	-	ND	-	-
3	Manure	-	-	-	C	-	-
1	Swab inside cat carcass	-	-	-	C	-	-
1	Cat hairs	-	-	-	C	-	-
1	Bones and skin	-	-	-	C	-	-
5	4 ceca of broilers	-	-	-	ND	-	-
5	2 ceca of laying hens	-	-	-	ND	-	-
6	Swab in young bull box	-	-	-	-	ND	-
4	Swab in cow box	-	-	-	-	ND	-
6	Boot swab in young bull box	-	-	-	-	ND	-
4	Boot swab in cow box	-	-	-	-	ND	-
3	Boot swab in the unwinter box	-	-	-	-	ND	-
1	Swab in mill and bushel	-	-	-	-	C	-
1	Swab inside empty wheat silo (wall and soil)		-	-	-	ND	-
1	Swab inside mill	-	-	-	-	-	ND
1	Swab inside the mill (top, axis and rotor)	-	-	-	-	-	ND
1	Swab on the top of the bushel	-	-	-	-	-	ND
1	Swab on the bottom of the bushel	-	-	-	-	-	ND
1	Swab inside the pipe at the exit of the silo	-	-	-	-	-	ND
1	Swab on the tubulars and grids from the mill		-	-	-	-	ND
1	Ashes from the burning site	-	-	-	-	-	ND

ND: Not detected; C: *C. botulinum* type C detected using Real-Time PCR; * samples were collected after cleaning and disinfection.

**Table 2 animals-09-01025-t002:** Detection of *C. botulinum* type C, D, C/D, and D/C using real-time PCR with two protocols for enrichment in samples collected on 7 symptomatic bovines.

Reference of the Animal and Samples	Protocol 1	Protocol 2
	(F-CMM, 70 °C 10 min)	TPGY
**Young bull no. 1**	-	-
Liver	ND	ND
Ruminal content	C	C
Rectum content	C	ND
Gall bladder	ND	ND
Hilum	ND	ND
Feces	C	ND
**Young bull no. 2**	-	-
Liver	C	C
Ruminal content	C	C
Rectum content	C	C
Gall bladder	C	C
Hilum	C	C
**Young bull no. 3**	-	-
Liver	C	C
Ruminal content	C	C
Rectum content	C	C
Gall bladder	C	ND
Hilum	C	C
Intestine	C	C
**Young bull no. 4**	-	-
Feces 1 *	C	ND
Feces 2 *	ND	ND
Liver	ND	C
Ruminal content	ND	ND
**Young bull no. 5 ^$^**	-	-
Feces 1 *	C	ND
Liver	ND	C
Ruminal content	ND	ND
**Young bull no. 6**	-	-
Feces 1 *	C	C
Feces 2 *	C	C
Liver	ND	C
Ruminal content	ND	C
**Cow no. 7**	-	-
Feces 1 *	ND	ND
Feces 2 *	C	ND
Liver	ND	C
Ruminal content	ND	C

Animal no. 1 was sampled immediately after euthanasia; animals no. 2 and 3 were sampled 24 h and 36 h after death; animals no. 4 to 7 recovered from symptoms but were eventually euthanized; * feces sample 1 was collected on 18 June and feces sample 2 on 25 June; ^$^ no feces available at the time of sampling for this animal; C: a signal for the gene encoding BoNT type C was detected; ND: no tested *bont* genes were detected.

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
