# Peer review of "A Case Report of a Botulism Outbreak in Beef Cattle Due to the Contamination of Wheat by a Roaming Cat Carcass: From the Suspicion to the Management of the Outbreak"

_animals, 2019, doi:10.3390/ani9121025_

Round 1
Reviewer 1 Report
This is an interesting paper relating to an outbreak of type C botulism on a farm. The paper is well written and clear.
1. The references should be carefully checked as some are incomplete or have errors
2. The 2 primers used, P1652 and P1653, are cited to reference 17, but I could not find them in that publication. A clear citation for these primers must be included.
3. Reference 25 is reported to contain information showing persistence for decades, but I could not find that information in the citation. Only a few years were covered in there. The authors should check this.
Reviewer 2 Report
This is an interesting case report regarding the outbreak of Botulism that occurred at Charolais cattle farm in France. The authors identified the sources of the contamination and detected strains of Botulism using real-time PCR with two protocols. The authors also describe the chronology of events using a schematic diagram which provides the reader with sufficient information regarding occurrence and measure taken after outbreaks. The manuscript is well written and can be accepted for publication in the present form.
Author Response
We would like to thank Reviewer 2 for the positive comments about our article.